# The staggered retreat of grounded ice in the Ross Sea, Antarctica since the LGM

Matthew A. Danielson[1], Philip J. Bart[1]

[1]Department of Geology and Geophysics, Louisiana State University, Howe-Russell Geoscience Complex E235, Baton Rouge, LA 70803, USA

*Correspondence to*: Matthew A. Danielson (mdani38@lsu.edu)

**Abstract.**

The retreat of the West Antarctic Ice Sheet (WAIS) in the Ross Sea after the LGM was more significant than for any other Antarctic sector. Here we combined the available chronology of retreat with new mapping of seismically resolvable grounding zone wedges (GZWs). Mapping GZWs is important because they record the locations of former stillstands in the extent of grounded ice for individual ice streams during the overall retreat. Our analysis shows that the longest stillstands occurred early in the deglacial and had millennial durations. Stillstands ended abruptly with retreat distances measured in the tens to hundreds of kilometers creating deep embayments in the extent of grounded ice across the Ross Sea. The location of embayments shifted through time. The available chronological data shows that cessation of WAIS and East Antarctic Ice Sheet (EAIS) stillstands were highly asynchronous across at least five thousand years. There was a general shift to shorter stillstands throughout the deglacial. Asynchronous collapse of individual catchments during the deglacial suggests that the Ross Sea sector would have contributed to multiple episodes of relatively small amplitude, sea-level rise as the WAIS and EAIS retreated from the region. The high sinuosity of the modern grounding zone in the Ross Sea suggests that this style of retreat persists.

# 1 Introduction

By the peak of the Last Glacial Maximum (LGM), grounded ice had advanced to the outermost continental shelf in the western Ross Sea, Antarctica, and to the continental shelf edge in the eastern Ross Sea (Anderson et al., 2014). The extent of grounded and floating ice was nearly as expansive as it could have been. Six fast flowing ice streams deeply eroded broad, foredeepened troughs across the continental shelf. Eroded sediment was transported in basal ice and/or subglacially (Alley et al., 2007; Alley et al., 1989; Powell et al., 1996; Prothro et al., 2018; Christoffersen et al., 2010). The sediment was

ultimately deposited either on the outer continental shelf or upper slope depocenters (Shipp et al., 1999). In the western Ross Sea, foredeepened Drygalski Trough (DT), JOIDES Basin (JB), and Pennell Trough (PT) extend to the continental shelf edge and were eroded during several successive glacial maxima. The outer parts of these troughs were partly backfilled with large-scale grounding zone wedges (GZWs) during and/or after the LGM. In the eastern Ross Sea troughs, where ice had reached the continental shelf edge, large trough-mouth fans were deposited on the upper slope (Mosola and Anderson, 2006).

During the post-LGM retreat, grounding line retreat paused within the outer reaches of the Glomar Challenger Basin (GCB), Whales Deep Basin (WDB) and Little America Basin (LAB), sufficiently long to deposit large GZWs (i.e., several tens of meters thick and tens of kilometers long) (Mosola and Anderson, 2006; Bart and Owolana, 2012) . The GZW sediment volumes partly reflect durations of grounding-line stillstands for individual ice streams (Bart and Cone, 2012; Bart et al., 2017; Bart and Owolana, 2012). Several previous studies have focused on the changing extent of grounded and floating ice

and timing of post-LGM retreat (Conway et al., 1999; Domack et al., 1999; Mosola and Anderson, 2006). Anderson et al. (2014) conducted the last Ross Sea synthesis of seismic stratigraphy and radiocarbon dates. More recently, Halberstadt et al. (2016) conducted a detailed evaluation of legacy multibeam data and identified GZWs and mega-scale glacial lineations associated with the LGM and post-LGM GZW stillstands. These stratigraphic data provide abundant evidence as to the progression of WAIS and EAIS retreat based on stratigraphic superposition. Here we build on the Halberstadt et al. (2016)

study of seafloor morphology by mapping the sediment volume of the GZWs across the six basins of the Ross Sea to evaluate the duration of individual grounding-zone stillstands. Establishing the former durations of GZWs is important to understand the regional scale ice sheet retreat in the Ross Sea and thus how ice-volume changes from Antarctica contributed

to global sea-level in the past. The paleo-perspective also informs our understanding of how additional contraction might proceed and contribute to future sea-level rise.

## 2 Methods

### 2.1 Regional seismic grid

Our study generated regional stratigraphic correlations of bounding surfaces across 22 surveys of multi- and single-channel reflection seismic data acquired from across the Ross Sea (Supplemental Table 1). The surveys include 510 seismic lines with a total coverage of ~54,000 km. The data are currently stored and maintained at the Antarctic seismic data library system (SDLS) at the Italian National Institute of Oceanography and Applied Geophysics (OGS) and have been used in previous studies such as Perez et al. (2021). Our mapping focused on GZWs interpreted to be of LGM and post-LGM ages, complementing the analyses of seafloor morphology by Halberstadt et al. (2016). The seismic profiles were interpreted in Petrel (Figure 1). The digital data used in this study have been processed prior to being archived at the SDLS. Individual seismic lines were imported as segy files into the software using separate files containing navigation data.

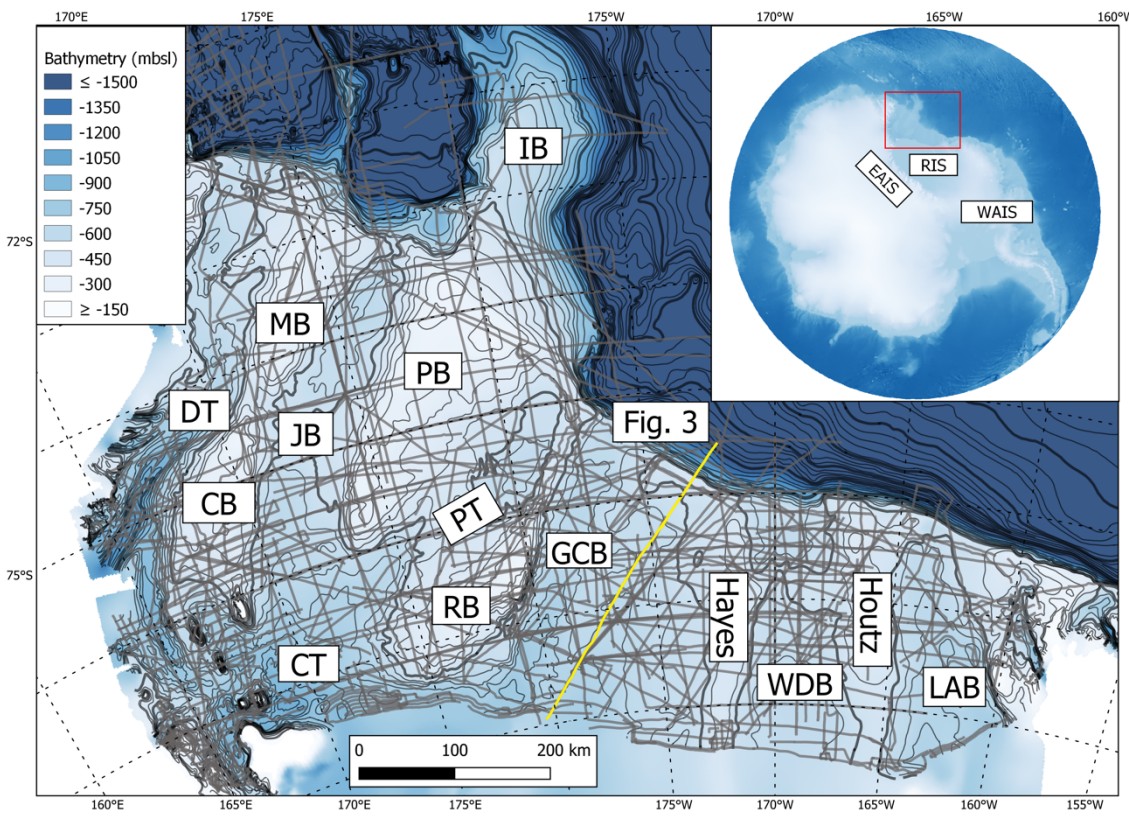


**Figure** 1**: Map of the Ross Sea showing seismic coverage. Bathymetry is from Davey and Nitsche (2013). Inset map of Antarctica using the International Bathymetric Chart of the Southern Ocean ice surface and bathymetry grid showing the locations of the East Antarctic Ice Sheet (EAIS) and the West Antarctic Ice Sheet (WAIS) in addition to the position of the Ross Ice Shelf (RIS) (Dorschel et al., 2022). Main map shows Ross Sea as indicated by the red box on the inset map. Labels of bathymetric troughs on**

**the Ross Sea shelf follow: LAB = Little America Basin, WDB = Whales Deep Basin, GCB = Glomar Challenger Basin, PT = Pennell Trough, CT = Central Trough, JB = JOIDES Basin, DT = Drygalski Trough. Labels of bathymetric banks on the Ross Sea shelf follow: Houtz = Houtz Bank, Hayes = Hayes Bank, RB = Ross Bank, PB = Pennell Bank, IB = Iselin Bank, MB = Mawson Bank, CB = Crary Bank. The position of the Figure 3 seismic profile is indicated with the yellow line.**

## 2.2 Seismic interpretation and isopach mapping of LGM and post-LGM GZWs

We focused on the seismically-resolvable LGM and post-LGM GZWs throughout the Ross Sea that have been identified in

previous seismic studies (Shipp et al., 1999; Mosola and Anderson, 2006; Bart and Owolana, 2012; Bart and De Santis,

2012; Bart et al., 2017). This includes GZWs identified in legacy multibeam data (Halberstadt et al., 2016). The seafloor

reflection and the unconformities bounding the top and base of the GZWs in each trough on the deglaciated Ross Sea shelf

were mapped using regional seismic stratigraphy and comparison to previous studies. Additional single channel paper

seismic lines from four surveys (NBP9307, NBP9401, NBP9501 and NBP9902) were used to supplement the interpretation

of major GZW features. These interpretations were completed on paper and then imported into Petrel using navigation files as a set of points in two-way travel time.

Two-way-travel time maps were made using convergent interpolation in Petrel with a cell size of 50 meters, where the computer-interpreted horizons were the primary input, and the paper-interpreted data were secondary input. Time-structure maps were made by subtracting the map of the GZW base from the seafloor map. Refraction sonobuoy measurements in the Ross Sea provide a regional record of sediment velocities (Cochrane et al., 1995; Cochrane et al., 1992). The points of the sonobuoy measurements taken from four expeditions were plotted in Petrel and interpolated to create depth and interval velocity maps. All the analysed GZW deposits were in shallow layers of sediment (upper 250 milliseconds) and thus only the uppermost interval velocity map was used. This section has velocities that vary from 1700 to 2200 meters/second across the region (Supplemental Figure 1). The interval velocity map was then used as an input to build a velocity model in Petrel. Time-structure maps were then depth-converted using the velocity model to create isopach maps for each GZW.

## 2.3 Volume and duration calculation

The isopach maps were used to calculate sediment volumes for the GZWs in QGIS software. The sediment volumes were then used as a basic parameter to estimate stillstand duration. The paleo-sediment flux for each of the paleo ice streams in the Ross Sea (Equation 1) is defined as $Q_s$. The paleo-sediment flux of an ice stream is the product of the paleo-drainage area (A) and average sediment yield (S) at the grounding zone where sediment is sourced from upstream subglacial erosion. Our estimates use a simple assumption of the sediment yield and paleo-drainage area. The sediment yield of $0.7 \pm 0.21$ mm yr$^{-1}$ derived by Bart and Tulaczyk (2020) for the WDB drainage area was applied to the adjacent catchment of LAB to infer a paleo-flux. The $0.7 \pm 0.21$ mm yr$^{-1}$ sediment yield was derived for the WDB middle shelf GZW which was determined to have been deposited during a stillstand whose onset and cessation dates are constrained by radiocarbon dates (Bart et al., 2018). For the western Ross Sea troughs of PT, JB and DT, we used a sediment yield of $0.49 \pm 0.21$ mm yr$^{-1}$ which is a value 30% less than the WDB value. This is due to the presence of crystalline and lithified sedimentary bedrock in western Ross Sea (Greenwood et al., 2021). A 30% lower sediment yield would be expected for a catchment floored by crystalline bedrock due to a higher resistance to erosion (Schlunegger et al., 2001). GCB received flow from both East and West Antarctica during the LGM (Licht et al., 2005). Thus, the sediment yield of $0.7 \pm 0.21$ mm yr$^{-1}$was used for the parts of the drainage

area from West Antarctica (Kamb, Whillans, Mercer ice streams) and the eastern Ross Sea shelf while the sediment yield of 0.49 ± 0.21 mm yr$^{-1}$ was used for the contribution from the outlet glaciers in East Antarctica such as Beardmore Glacier (Figure 2).

$$Q_s = AS \qquad (1)$$

Paleo-drainage areas were estimated for each of the paleo-troughs of the Ross Sea using the drainage area of the present-day WAIS ice-streams and EAIS outlet glaciers and projecting their extents into each of the troughs on the outer continental shelf (Figure 2). The terminus of each paleo-drainage area is the seaward edge of its respective grounding zone wedge. The approach assumes single-ice stream capture for WDB and LAB. The GCB received drainage from the combination of Kamb, Whillans and Mercer ice streams based on the sub-ice shelf topography shown in the ROSETTA project in addition to other

East Antarctic glaciers (Tinto et al., 2019; Licht et al., 2005). The JB and PT shared capture from Byrd and other smaller East Antarctic catchments. The upstream drainage area shared by JB and PT was halved for calculations to reflect the shared source of sediment. DT primarily received ice flow from David Glacier during the LGM (Licht and Palmer, 2013; Licht et al., 2014). There was a reorganization in flow in the southern region of western Ross Sea during the retreat that is marked by backstepping geomorphological evidence towards David Glacier (Greenwood et al., 2018). Thus, the paleo-flux for the

GZW mapped within the interior middle shelf of JB nearest to Franklin Island (Figure 4 m; and Table 1) was calculated using a modified drainage area from Mawson, Mackay and David Glaciers with no input from Byrd Glacier (Figure 2). Grounding duration at each location was calculated using the method following Bart and Tulaczyk (2020) where ΔT is the grounding duration, V is the total volume of GZW sediment and $Q_s$ is the paleo-sediment flux (Equation 2).

$$\Delta T = \frac{V}{Q_s} \qquad (2)$$

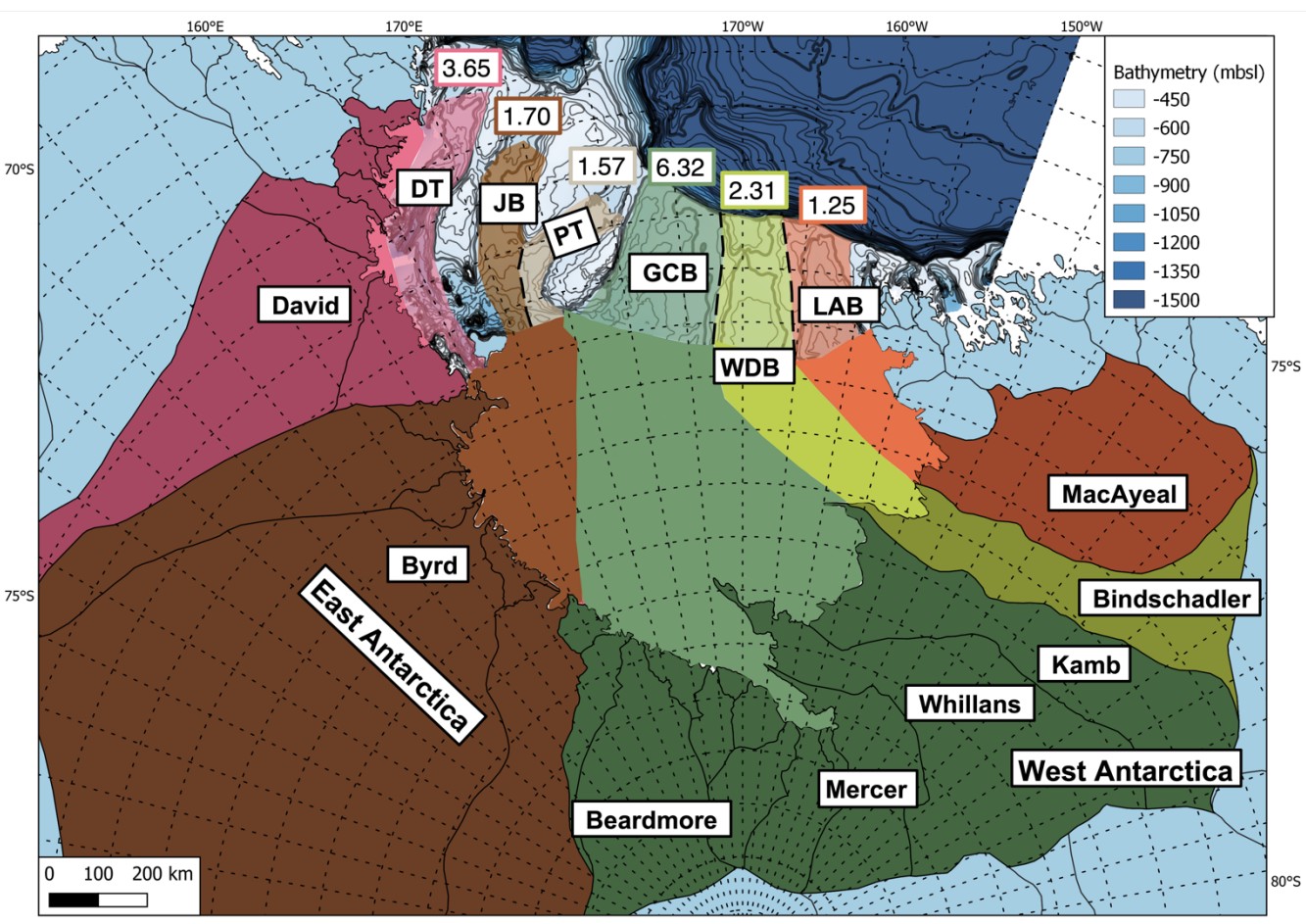


**Figure 2: Estimated paleo-drainage catchments for each of the major troughs across the Ross Sea during the LGM. The darker shades correspond with the present-day drainage areas while the lighter shades correspond with the projected paleo-drainage into the troughs when the ice sheet extent was expanded. orange = LAB, yellow= WDB, green = GCB, brown = Pennell and JOIDES, pink = Drygalski Trough. The heavy black dashed line shows the separation between paleo-drainage of adjacent troughs.**

**Drygalski Trough, JOIDES Basin and Pennell Trough capture drainage from East Antarctic glaciers. The upstream drainage area for JOIDES Basin and Pennell Trough is shared with the area upstream of dashed black flow divide being divided by two for the calculations. GCB drained a combination of WAIS paleo-ice streams and EAIS outlet glacier flow. LAB and WDB received sediment from individual WAIS paleo-ice streams. LGM paleo-drainage areas are labeled for each trough where each label is shown in the form of $10^5$ km². Modern drainage area polygons are defined by IMBIE 2016 (Mouginot et al., 2017; Rignot et al.,**

**2013).**

## 3 Results

### 3.1 Seismic-resolvable GZWs on the Ross Sea shelf

Stratigraphic correlations on the regional seismic transects yielded two-way travel time maps for the Ross Sea outer continental shelf and middle continental shelf (Supplemental Figure 2). The inner continental shelf is covered by the Ross

Ice Shelf and hence cannot be investigated using the marine seismic profiles used in this study. Mapped horizons bound GZWs from the base of the LGM unconformity to the seafloor (Figure 3). Seventeen GZWs were identified and mapped within the Ross Sea trough basins (Appendix 1; Supplemental Table 3). Fourteen of these GZWs (Figure 4 a-h, k-m, o-q and Table 1) have been identified from previous studies (e.g., Anderson et al., 2014; Halberstadt et al., 2016). Three new GZWs were mapped from the regional seismic data in the inner reaches of the middle continental shelf sectors of JB, PT and GCB (Figure 4 i-j, n; and Table 1). The WDB and PT have two GZWs at the shelf edge and on the middle continental shelf. JB has a GZW at the middle continental shelf and two proximal to the modern ice shelf calving front in the inner reaches of the trough. The GCB, which has the largest drainage area of all the paleo ice streams, has five GZWs with one at the shelf-edge, one on the middle-shelf and three in the inner reaches of the trough proximal to the modern ice shelf calving front (Table 2; Figure 4).

In the eastern Ross Sea, the shelf edge and outer continental shelf GZWs define part of the modern banks. In the western Ross Sea, the GZWs are trough-confined except for the inner reaches of the middle continental shelf GZWs in DB that were deposited on the banks adjacent to the foredeepened section of the trough (Baroni et al., 2022). Time structure maps were generated for the top and base of each GZW and depth converted. The difference of these two surfaces gives thickness maps for the seventeen GZWs in the Ross Sea (Figure 4).

## 3.2 Grounding durations of Ross Sea GZWs

Sediment volumes for each GZW are shown in Table 1. The outer shelf GZWs in WDB, GCB, JB and DT are the largest with sediment volumes on the order of $10^3$ km$^3$. The inner reaches of the middle continental shelf GZWs have the smallest volumes on the order of $10^2$ km$^3$. The drainage area for the ice streams that deposited these GZWs includes the projected paleo-ice stream drainage pathways (Figure 2) up to the topset-foreset boundary of the mapped GZW. Paleo-flux values are estimated from the product of drainage area with the sediment yield. Stillstand duration was then calculated from the paleo-sediment fluxes and GZW sediment volumes (Tables 2 and 3).

The JB outer continental shelf GZW has the longest duration of ~5.0 kyrs. The inner reaches of the middle continental shelf GZWs in GCB have the shortest durations on the order of $10^1$ - $10^2$ years. Durations calculations include an uncertainty of $\pm$ 2

milliseconds from uncertainty in the TWTT measurement of the GZWs as well as a ± 50 m/s uncertainty from the velocity

model used to convert the TWTT maps to depth.

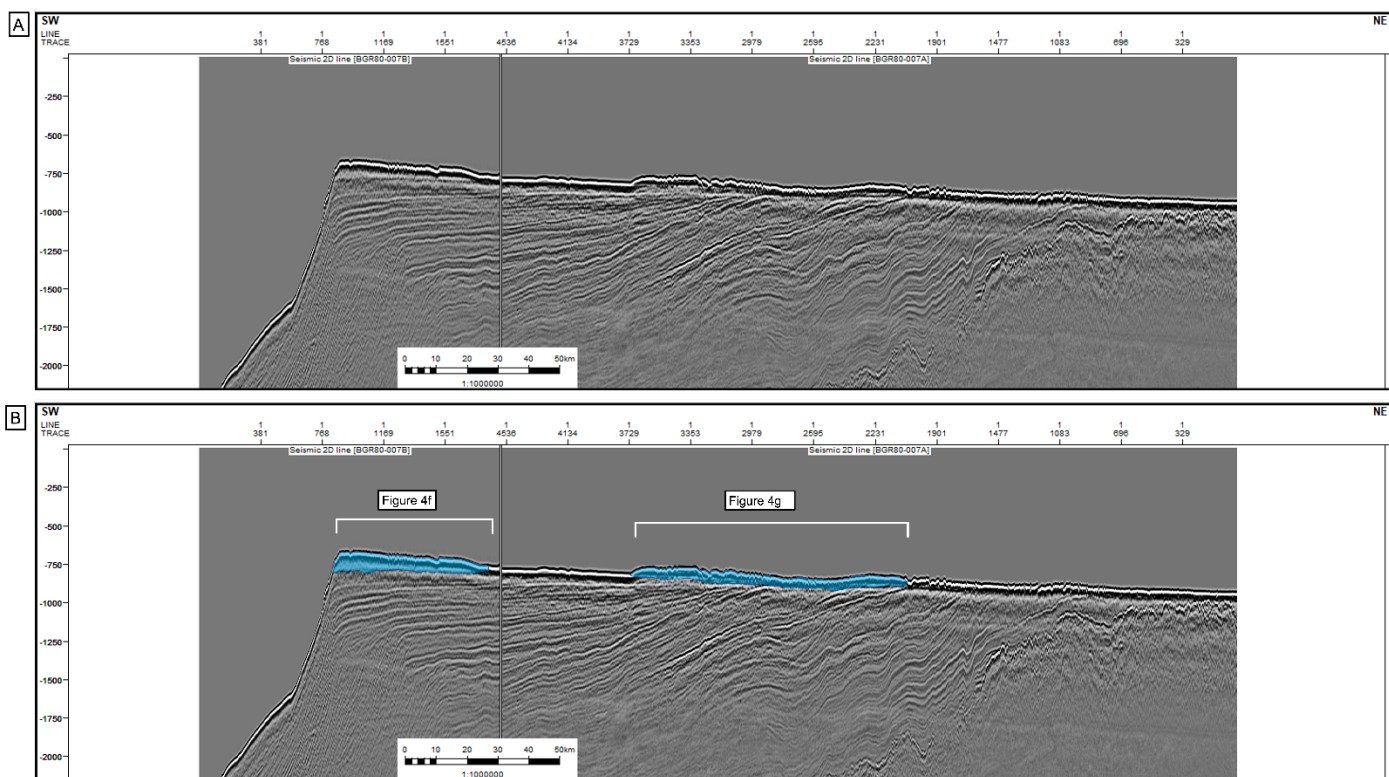

**Figure 3: Uninterpreted (A) and interpreted (B) seismic line BGR80-007 through GCB. Location of line is indicated with Figure 1.**
**Blue shaded region shows interpreted GZWs. Position of GZWs and thickness map are shown in Figure 4.**

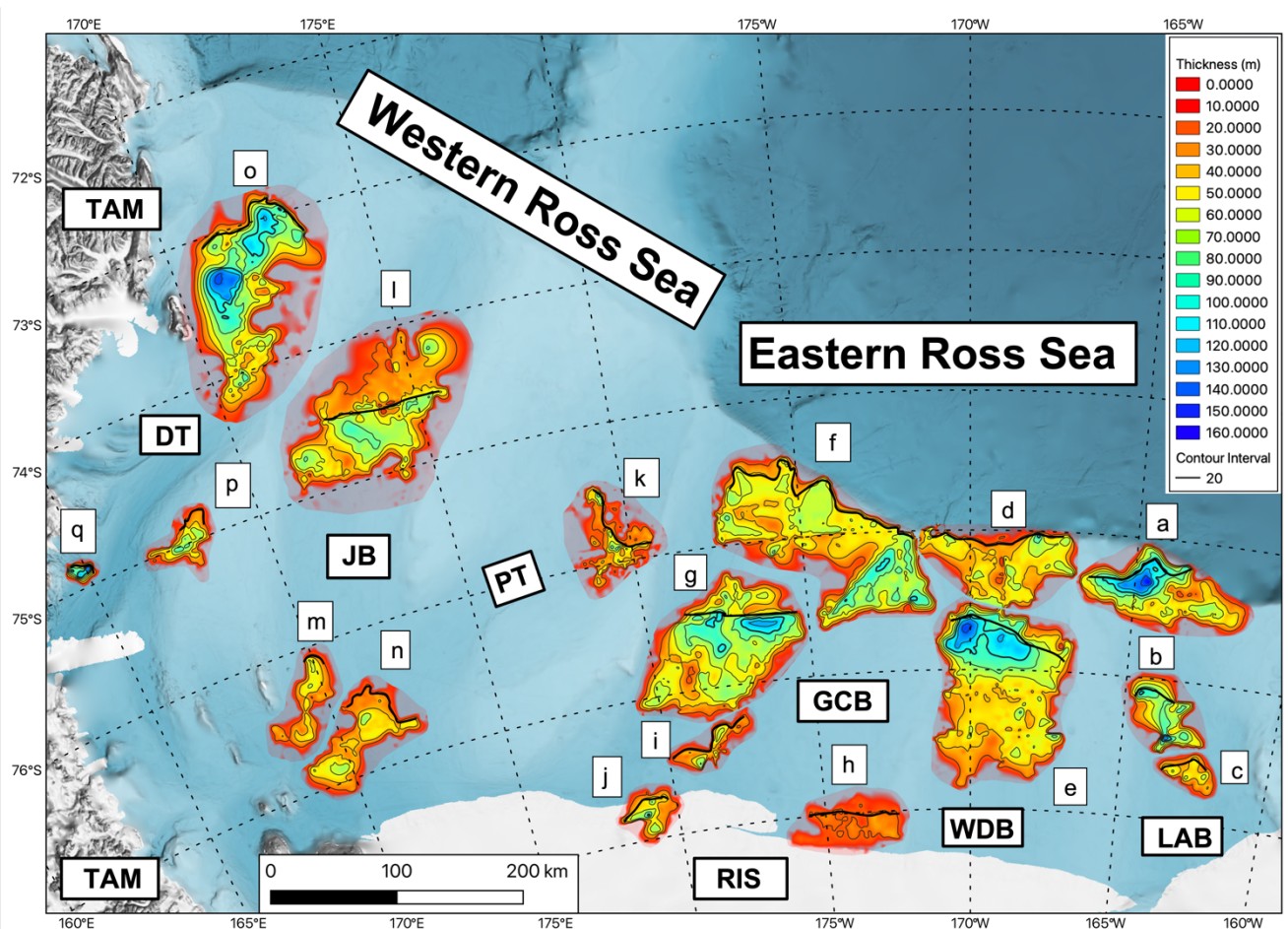

**Figure 4: Thicknesses contour maps of Ross Sea GZWs. The heavy black line shows the approximate location of boundary between the GZW topset and foreset. GZWs are labeled "a" to "q" starting in the east with LAB outer shelf GZW. Letter labels of GZWs correspond with records in Table 1. Labels of geometric features follow: LAB = Little America Basin, WDB = Whales Deep Basin, GCB = Glomar Challenger Basin, PT = Pennell Trough, CT = Central Trough, JB = JOIDES Basin, DT = Drygalski Trough, TAM = Transantarctic Mountains, RIS = Ross Ice Shelf. Bathymetry shown in the base map is from IBSCO v2 (Dorschel et al., 2022).. Contour interval = 20 m**

| GZW Wedge Location | Estimated Paleo Drainage Area, $A$ ($10^5$ km$^2$) | GZW Volume, $V$ (km$^3$) | Paleo Sediment flux, $Q$ ($10^8$ m$^3$ a$^{-1}$) | Duration, $\Delta T$ (yr) |
|---|---|---|---|---|
| a. LAB Outer Shelf | 1.25 | $301 \pm 19$ | $0.87 \pm 0.26$ | $3445 \pm 1136$ |
| b. LAB Middle Shelf | 1.11 | $120 \pm 8$ | $0.79 \pm 0.23$ | $1535 \pm 506$ |
| c. LAB Middle Shelf Inner Reaches | 1.05 | $33 \pm 3$ | $0.74 \pm 0.22$ | $453 \pm 149$ |
| d. WDB Outer Shelf | 2.31 | $216 \pm 16$ | $1.62 \pm 0.49$ | $1027 \pm 419$ |
| e. WDB Middle Shelf | 2.23 | $534 \pm 35$ | $1.67 \pm 0.47$ | $3200 \pm 700$ |
| f. GCB Outer Shelf | 6.32 | $610 \pm 41$ | $4.10 \pm 1.33$ | $1487 \pm 455$ |
| g. GCB Middle Shelf West | 6.17 | $523 \pm 33$ | $4.00 \pm 1.30$ | $1308 \pm 399$ |
| h. GCB Inner Reaches East | 5.80 | $47 \pm 7$ | $3.74 \pm 1.22$ | $126 \pm 38$ |
| i. GCB Inner Reaches West | 5.93 | $33 \pm 4$ | $3.83 \pm 1.24$ | $87 \pm 27$ |
| j. GCB Inner Reaches Ross Bank | 5.72 | $45 \pm 4$ | $3.68 \pm 1.20$ | $121 \pm 37$ |
| k. PT Middle Shelf | 1.57 | $69 \pm 8$ | $0.77 \pm 0.33$ | $898 \pm 207$ |
| l. JOIDES Middle Shelf | 1.70 | $421 \pm 37$ | $0.83 \pm 0.36$ | $5072 \pm 1170$ |
| m. JOIDES Inner Reaches 1 | 0.90 | $58 \pm 6$ | $0.44 \pm 0.19$ | $1340 \pm 309$ |
| n. JOIDES Inner Reaches 2 | 1.48 | $117 \pm 11$ | $0.73 \pm 0.31$ | $1612 \pm 372$ |
| o. DT Outer Shelf | 3.65 | $583 \pm 21$ | $1.79 \pm 0.77$ | $3257 \pm 752$ |
| p. DT Inner Reaches 1 | 1.17 | $48 \pm 5$ | $0.57 \pm 0.25$ | $844 \pm 195$ |
| q. DT Inner Reaches 2 | 1.17 | $20 \pm 1$ | $0.57 \pm 0.25$ | $350 \pm 81$ |

**Table 1: Summary table of GZWs shown in Figure 4 with drainage area, volume, paleo sediment flux and duration with respective uncertainties.**


| | Little America | Whales Deep | Glomar Challenger | Pennell | JOIDES | Drygalski |
|---|---|---|---|---|---|---|
| Drainage area ($10^5\,km^2$) | 1.25 | 2.31 | 6.32 | 1.57 | 1.70 | 3.65 |
| LGM Paleo-sediment flux ($10^8\,m^3\,a^{-1}$) | 0.87 ± 0.26 | 1.62 ± 0.49 | 4.10 ± 1.33 | 0.77 ± 0.33 | 0.83 ± 0.36 | 1.79 ± 0.77 |

**Table 2: Drainage areas and paleo-flux of LGM positions with respective uncertainties.**

| Shelf position | Little America | Whales Deep | Glomar Challenger | Pennell | JOIDES | Drygalski |
|---|---|---|---|---|---|---|
| 1. **outer continental shelf** | 301 ± 19 | 216 ± 16 | 610 ± 41 | | | 583 ± 21 |
| 2. **middle continental shelf** | 120 ± 8 | 534 ± 35 | 523 ± 33 | 69 ± 8 | 421 ± 37 | |
| 3. inner **middle continental shelf** | 33 ± 3 | | 47 ± 7, 45 ± 4, 33 ± 4 | | 58 ± 6, 117 ± 11 | 48 ± 5, 20 ± 1 |

**Table 3: GZW Volumes ($km^3$) arranged by Ross Sea shelf position with uncertainty.**



## 4 Discussion

### 4.1 Stillstand Durations on the Ross Sea continental shelf

#### 4.1.1 Millennial scale stillstand durations on the outer and middle continental shelf

We present durations estimated using the sediment yield from the WDB middle continental shelf stillstand for the eastern Ross Sea troughs and a yield that is 30% less for the western Ross Sea troughs. The durations for GCB were calculated using both yields to reflect the drainage from both East and West Antarctica. The calculated durations suggest that the largest GZWs in the Ross Sea had stillstands lasting up to a few millennia (Table 4). This general assessment is strongly supported by radiocarbon dates using benthic foraminifera collected in till of the WDB middle continental shelf stillstand (Bart et al., 2018). The dates suggest that the stillstand had begun by $14.7 \pm 0.4$ cal kyr BP before retreating by $11.5 \pm 0.3$ cal kyr BP. The GZWs on the Ross Sea shelf are generally larger than those on other Antarctic continental shelves (Batchelor and Dowdeswell, 2015). In the eastern Ross Sea troughs, larger sediment volumes are partly related to ice stream erosion across the broad West Antarctic catchment areas much of which is underlain by sedimentary bedrock (Tinto et al., 2019). In the Western Ross Sea troughs, there was expanded flow from East Antarctic glaciers through the Transantarctic Mountains during the LGM that provided sediment flux (Licht et al., 2005; Licht and Palmer, 2013). Exposed outcrops in the Transantarctic Mountains suggest that the bedrock that underlaid the expanded flow was primarily crystalline bedrock and metasediments (Li et al., 2020). The high sediment flux and widespread sediment aggradation at the grounding lines (Table 3) would have also contributed to long stillstands by countering the effect of ice-stream thinning associated with the deglaciation as flow accelerates, sea-level rises and global climates warm (Anandakrishnan et al., 2007).

#### 4.1.2 Variable stillstand durations between troughs

Our data suggest that the grounding line stillstands on the Ross Sea shelf were of millennial and centennial durations. We focus on comparisons to WDB middle continental shelf stillstand because its duration is constrained by radiocarbon dates (Bart et al., 2018). In map view, the WDB middle continental shelf appears to be in regional alignment to the outer continental shelf GZWs in DT and JB, and the middle continental shelf GZWs in PT, GCB, and LAB (Figure 4). Comparison between the middle continental shelf stillstand durations shows that the PT, LAB and GCB stillstand durations are shorter than the WDB stillstand duration. The lower durations for the GCB are a result of a higher paleo-drainage area

despite the similar volumes. The outer continental shelf stillstand in DT has a comparable duration to the WDB stillstand. The middle continental shelf stillstand for the JB has the longest duration of ~5 kyrs.


| Shelf position | Little America | Whales Deep | Glomar Challenger | Pennell | JOIDES | Drygalski |
|---|---|---|---|---|---|---|
| 1. outer continental shelf | 3445 ± 1136 | 1027 ± 419 | 1487 ± 455 | | | 3257 ± 752 |
| 2. middle continental shelf | 1535 ± 506 | 3200 ± 700 | 1308 ± 399 | 898 ± 207 | 5072 ± 1170 | |
| 3. inner middle continental shelf | 453 ± 149 | | 126 ± 38, 121 ± 37, 87 ± 27 | | 1340 ± 309, 1612 ± 372 | 844 ± 195, 350 ± 81 |

**Table 4. GZW Durations arranged by Ross Sea shelf position with uncertainty. All durations are presented in years.**


### 4.1.3 Stillstand durations within individual troughs

By stratigraphic superposition, GZWs on the OCS are older than those on the middle continental shelf. Within those basins with more than one GZW, e.g., LAB, our data suggest significant reductions in stillstand durations as the deglacial progressed. The shift to shorter stillstands on the inner reaches of the middle continental shelf is generally consistent with

tenets of the marine-ice-sheet-instability hypothesis which predicts unstable grounding line retreat across the foredeepened continental shelf (Weertman, 1974; Joughin and Alley, 2011). After the deposition of a GZW in the interior of JB and PT (Figure 4 n; and Table 1), there was a reorganization in the flow to primary input from Mawson and Mackay Glaciers (Greenwood et al., 2018). Thus, the smaller volume GZW deposited near Franklin Island (Figure 4 m; and Table 1) has a similar duration to the previous stillstand position. Grounding zone deposits that are too small or thin to map with seismic

data are reported from several of the Ross Sea shelf troughs from high-resolution swath bathymetry (Halberstadt et al., 2016; Simkins et al., 2017; Greenwood et al., 2018; Bart and Kratochvil, 2022). We follow other studies that suggest these small-

scale features would logically correspond to decadal and/or annual timeframes (Livingstone et al., 2016; Dowdeswell et al., 2019).

## 4.2 Post-LGM erosion rates in the Ross Sea

A key assumption of our study is that erosion rates ranged from $0.7 \pm 0.21$ to $0.49 \pm 0.21$ mm yr$^{-1}$ for West and East Antarctic catchments respectively. This relatively broad range overlaps with the erosion rate estimates for a modern WAIS ice stream (Alley et al., 1986, 1987). The yields are also within the range of erosion for land-based glaciers from Norway, Svalbard and Switzerland and upper-slope Bear Island trough mouth fan depocenters (Elverhøi et al., 1998). These and other studies show that yield is affected by the degree of ice cover, regional climate, and associated precipitation, and

presence/absence of meltwater. All the Ross Sea catchments are south of 70°S and over the post-LGM timeframes we considered, the climates were uniformly colder than present (Cuffey et al., 2016). The catchments were all entirely covered by grounded ice so the degree of glaciation could not have been a significant contributor to erosion rate differences between drainage areas. There is no evidence of warmer-than-present intervals that might have significantly increased meltwater production that would have contributed to high end erosion rates (Cuffey et al., 2016). The lowest erosion rates are expected

for large catchments with slow-flowing cold ice (Hallet et al., 1996). Deglacial erosion rates are expected to be high because of the rapid flow of warmer ice (Kingslake et al., 2018; Koppes and Montgomery, 2009).

Additional controls on erosion rates and stillstand durations are subglacial topography, subglacial geology and external atmospheric or oceanographic forcing. Topographic controls on ice stream flow include bottleneck effects from a cross-sectional constriction of trough as well as localized highs of the seafloor (Dowdeswell and Fugelli, 2012; Mckenzie et al.,

2023; Danielson and Bart, 2019). Variations in subglacial geology can also impact erosion rates and stillstand durations where less erodible crystalline and indurated sedimentary bedrock can facilitate longer duration grounding zone deposition (Klages et al., 2015). Bedrock outcropping at the seafloor can decelerate ice sheet retreat and trigger stillstands (Klages et al., 2014). For the GZWs mapped in this study, there was no presence of outcropping bedrock. External climatic forcing is an important control on grounding line stability. Model results suggest that different ocean and atmosphere forcing

combinations in the early deglacial are important for controlling the timing and pattern of retreat (Lowry et al., 2020).

Changes in the subglacial topography, substrate type, or external climatic forcing could have contributed to variations in erosion rates across the Ross Sea troughs and is reflected in our uncertainty estimates.

All the West Antarctic catchment area is underlain by unconsolidated sediments and sedimentary strata save for small areas of exposed basement (Wilson and Luyendyk, 2006; Jordan et al., 2020; Anderson and Bartek, 1992). Substrates are expected to have similar erodibilities. Thus, the sediment yield of $0.7 \pm 0.21$ mm yr$^{-1}$ derived by Bart and Tulaczyk (2020) for the WDB drainage area is most appropriate for the eastern Ross Sea troughs of LAB and WDB. The East Antarctic parts of the Ross Sea catchments are underlain by less erodible crystalline rocks and lithified sedimentary bedrock (Greenwood et al., 2021). Yields from basement rock are lower by 30% compared to sedimentary strata (Schlunegger et al., 2001). The sediment yield of $0.49 \pm 0.21$ mm yr$^{-1}$ is then most appropriate for the western Ross Sea troughs of PT, JB and DT where a less erodible substrate would produce lower average sediment fluxes.

### 4.3 A staggered post-LGM retreat of grounding lines in the Ross Sea

Grounding line retreat from the DT outer continental shelf stillstand is estimated to have occurred at 16.5 cal kyr BP (Prothro et al., 2020; Anderson et al., 2014; Cunningham et al., 1999). Prothro et al. (2020) used benthic foraminifera from glacial proximal sediments to show that middle shelf grounding zone stillstands in the JB and PT ended at 15.1 cal kyr BP and 13.3 cal kyr BP respectively. Radiocarbon dates from the WDB show that ice had retreated from the shelf edge by $14.7 \pm 0.3$ cal kyr BP and that retreat from the middle continental shelf occurred at $11.5 \pm 0.3$ cal kyr BP (Bart et al., 2018). Bart and Cone (2012) proposed the GCB stillstand ended at 27.5 cal kyr BP. A pre-LGM retreat is precluded because data presented by Halberstadt et al. (2016) require that ice remained grounded in both GCB and LAB until after a grounding line embayment opened in the WDB at 11.5 cal kyr BP (Bart and Kratochvil, 2022). The oldest date from deglacial sediment overlying the foreset of the middle continental shelf GZW in GCB requires that the stillstand had ended by 8.7 cal kyr BP (Bart and Cone, 2012). We apply the same age of retreat (8.7 cal kyr) to the LAB middle continental shelf stillstand because the only other radiocarbon ages are from core tops.

Our data do not support previous studies that suggested that retreat occurred in a gradual lockstep fashion (Conway et al., 1999). Instead, both the chronology and stillstand duration data suggest that grounding line retreat proceeded in an unsteady episodic retreat style within individual troughs (Table 5 and Figure 5). The earliest retreat in DT may be partly related to the

greater depth of the DT. The subsequent opening of an embayment in PT may have been related to its small catchment area that delivered relatively low volumes of ice to the grounding zone. The sustained grounding in the JB may have been associated with both its larger catchment, flow capture from the PT catchment and buttressing from its adjacent broad shallow banks. The long stillstand duration in the WDB may have been aided by antecedent topography that includes a

bottleneck constriction at the location of the middle continental shelf grounding stillstand (Danielson and Bart, 2019) plus the apparent rapid sediment aggradation following ice shelf break up at 12.3 cal kyr BP (Bart and Tulaczyk, 2020). The available age control (see above) suggests that up to three millennia may have elapsed before grounded ice retreated from the GCB and LABs but here the chronologies are poorly constrained.

We acknowledge that the retreat chronology is likely to change as more radiocarbon data are generated. With the available

constraints, our data supports other previous studies that suggested that retreat was not synchronous or in lock step between troughs (Halberstadt et al., 2016; Prothro et al., 2020; Mosola and Anderson, 2006). Neither the onset, duration or termination of Ross Sea stillstands appear to be related to global or regional scale forcing mechanism with the possible exception of the WDB middle continental shelf stillstand which may be bracketed between intervals of rapid, large amplitude sea level rise at meltwater pulses (MWPs) 1a and 1b (Lin et al., 2021). These data are not consistent with the view

that WAIS and EAIS contraction in the Ross Sea contributed significantly to the sustained sea-level rise during either MWP1a or 1b. An asynchronous opening of grounding-line embayments would have been associated with multiple episodes of short-lived accelerated sea-level rise. The marked sinuosity of the modern grounding line in the Ross Sea suggests that this staggered retreat persists through to present.


| GZW Wedge Location | Retreat mode Duration, $\Delta T$ (yr) | Nearest retreat date (cal yr BP) | Grounding start date (cal yr BP) | Date Reference |
|---|---|---|---|---|
| b. LAB Middle Shelf | 1535 | $8715_b \pm 70$ | $10250 \pm 70$ | NBP0802 PC2 7 - 9 cm (Bart and Cone, 2012) |
| e. WDB Middle Shelf | 3200 | $11500_b \pm 300$ | $14701 \pm 300$ | NBP1502B KC07 (Bart et al., 2018) |
| g. GCB Middle Shelf West | 1308 | $8715_b \pm 50$ | $10023 \pm 50$ | NBP0802 PC2 7 - 9 cm (Bart and Cone, 2012) |
| k. PT Middle Shelf | 898 | $15121_b \pm 270$ | $16019 \pm 270$ | NBP1502A KC17 144 – 145 cm (Prothro et al., 2020) |
| l. JOIDES Middle Shelf | 5072 | $13315_a \pm 240$ | $18387 \pm 240$ | NBP1502A KC48 Prothro et al., 2020 |
| o. DT Outer Shelf | 3257 | $16519_b \pm 260$ | $19776 \pm 260$ | NBP9501 KC37 (Prothro et al., 2020; Anderson 2014; Cunningham et al., 1999) |


a: AIO bulk sediment date

b: Benthic carbonate material from grounding zone sedimentation

**Table 5: Grounding start date model for the middle continental shelf GZWs.**

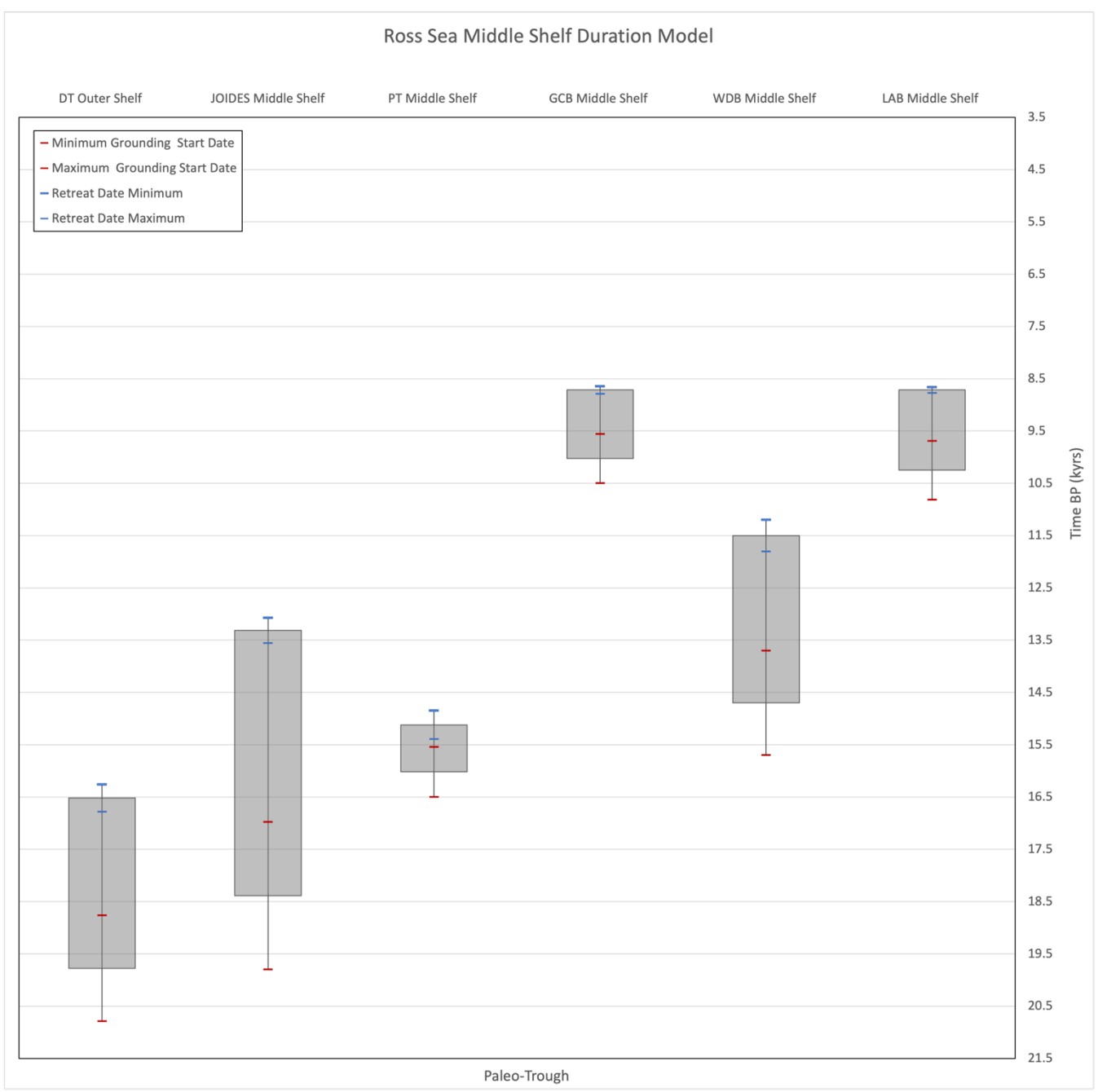


**Figure 5. Duration age model for the middle continental shelf GZWs in the Ross Sea using the durations and nearest retreat date plotted in Table 5. The grey box shows the median grounding zone retreat date added to the estimated duration to yield a grounding start date. Blue markers show the uncertainty of the retreat date. Red markers show the uncertainty of the retreat date**

**added to the duration yielding minimum and maximum grounding start dates. The uncertainty of the duration estimates is**

**incorporated in the minimum and maximum start dates.**

**5 Conclusion**

Given the inherent uncertainties in our approach, we acknowledge that future work should focus on more directly constraining onset, duration, and cessation of grounding zone stillstands with radiocarbon data. With the available

chronologic data, seismic mapping of GZWs provides reasonable first-order estimates for stillstand duration. The locations and sediment volumes of GZWs suggest millennial to centennial duration stillstands for Ross Sea ice streams during the early phases of the post-LGM retreat followed by a shift to significantly shorter stillstands. Combined with the available age control, our first-order duration estimates strongly suggest a staggered retreat that formed deep grounding-line embayments between troughs. These results can be used as inputs to ice sheet models to better constrain contributions to the post-LGM

sea-level rise as the deglacial progressed in the Ross Sea. Asynchronous collapse of individual catchments occurring over the course of the post-LGM suggests that the Ross Sea sector contributed to multiple episodes of relatively small-amplitude sea-level rise rather than fewer intervals of rapid large-amplitude sea level changes from a regionally synchronous retreat. The high sinuosity of the modern grounding zone in Ross Sea suggests that this retreat style persists.

**Data Availability**

The data that support the findings of this study are openly available in the SDLS hosted at OGS at https://sdls.ogs.trieste.it/cache/index.jsp. A full list of seismic surveys used in this study are listed in Supplemental Table 1.

**Author contribution**

MD performed the mapping of the seismic data and figure generation; MD and PB interpreted the results.; MD and PB wrote and edited the manuscript.

**Competing interests**

The authors declare that they have no conflict of interest.


**Acknowledgments**

Support for the project was provided by a United States National Science Foundation Office of Polar Programs Antarctic

Earth Sciences Division grant (#1841136) to Bart. Seismic data used for this project were accessed from the Antarctic

Seismic Data Library System (SDLS) hosted at the National Institute of Oceanography and Applied Geophysics (OGS). We

thank the original collectors of these data. A full list of the seismic surveys used can be found in supplemental table 1.

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
