# Peer review of "The staggered retreat of grounded ice in the Ross Sea, Antarctica since the LGM"

_EGUsphere, 2023_

## Author Response (AR1)

*Note: The reviewer comments are presented as previously posted in normal black font. Our responses appear below each reviewer comment italicized. All line numbers listed in responses to comments refer to the version with accepted changes.*

**#1 Referee comment for preprint manuscript egusphere-2023-1397 "The staggered retreat of grounded ice in Ross Sea, Antarctica since the LGM" by Matthew A. Danielson and Philip J. Bart**

**General comments**

*Danielson and Bart* present an interesting compilation of seismic data from the Ross Sea Embayment (RSE) shelf to 1) assess the retreat style of grounding lines during and subsequent to the Last Glacial Maximum (LGM) and to 2) estimate grounding line stillstand durations during general retreat represented by several grounding-zone wedges (GZWs) situated in different paleo-ice stream troughs. It is a well written and a concise presentation of available data that the authors try to apply to answer longstanding research questions related to the past grounding line behavior within major high-latitude glacial outlets, i.e., which stillstand durations do individual grounding-zone wedges represent and how can that knowledge be used to characterize past ice sheet retreat in large embayments such as the Ross Sea Embayment. However, in the current manuscript version too many general assumptions regarding accumulation and erosion rates are made that base on a single data set of reliable radiocarbon age constraints that made actual calculations for one grounding line stillstand event in the eastern Ross Sea Embayment possible (Bart et al., 2018; Bart and Tulaczyk, 2020). The authors themselves highlight the apparent modern asynchronicity of grounding line response within individual glacial troughs in the Ross Sea sector to external forcing, which can therefore also be expected to grounding line retreat in the past. Generalizing accumulation and erosion rates across individual glacial troughs is therefore quite a risky and speculative approach. Therefore, the author's key assumptions need a more robust justification since quite far-reaching conclusions are drawn from this. In the following specific comments, I elaborate on this in more detail and really hope that the authors will be able to provide conclusive reply to this, i.e., discuss those issues in more detail in their manuscript, since I generally think that this work will deliver significant broader insight into past grounding-line dynamics in a large Antarctic embayment. This will not only be important to better understand past Antarctic ice sheet dynamics but will also deliver significant information for numerical modelers that aim to improve simulations for the grounding line's future response.

**Specific comments**

Major

*Note: The reviewer comments are presented as previously posted in normal black font. Our responses appear below each reviewer comment italicized. All line numbers listed in responses to comments refer to the version with accepted changes.*

- Chapter 2.3 "Volume and duration calculation" and resulting discussion (Chapters 4.1, 4.2, and 4.3): As 2.3 is the most important paragraph of your method section, you need to justify your approach more clearly and carefully here. Transferring the accumulation and erosion rates from one age-constrained GZW to other major GZWs – particularly from different glacial troughs – is very risky and quite speculative. Consider that large outer shelf GZWs accumulated during initial post-LGM retreat, a time of rapid GL retreat along the Antarctic margin that may have involved large amounts of subglacial meltwater that elevated sediment output at the grounding zone. Further, the initial forcing on GL retreat has likely been quite different along the Ross Sea sector (Lowry et al., 2019; Sci Adv 5), resulting in heterogenous responses of GLs and GZW formation across different glacial troughs in the RSE. In lines 217–219 you specify that "the degree of glaciation could not have been a significant contributor to erosion rate differences between drainage areas" and that "there is no evidence of warmer-than-present intervals that might have significantly increased meltwater production that would have contributed to high end erosion rates" but not only the degree of glaciation affects erosion rates in a glacial trough but also different topographic settings, subglacial geology, and exposure to external climatic and oceanic forcing. The outermost parts of the continental shelf are usually characterized by very low ––gradients that are not only a lot more susceptible to minor forcing but also result in larger grounding zones of thinner ice that has probably been in constant motion by e.g., tidal movements and wave action. So, those environments cannot not really be compared to modern grounding zones that may well be susceptible to a warmer climate but are usually a lot further inland on higher gradient beds mostly made from more resistant bedrock. And yes, West Antarctica is probably mostly made from sedimentary rocks but for most locations – particularly bene ath the ice sheet – we simply don't know. And even if sedimentary rocks dominate WAIS' substrate, those may be metasedimentary in places (e.g., Jordan et al., 2020, Nat Revs Earth & Env), i.e.,

still be of very high resistance against glacial erosion. In my opinion, this should be discussed and considered in the manuscript as it may have affected past GL retreat by quite a lot and may have been the reason for stillstands in places as it can largely affect the bed gradient and thus GL retreat across those sections. Generally, there is a strong need to discuss the high uncertainties related to your approach outlined in 2.3. Also, elaborate more on the sediment yield and erosion rate – as of now it reads if the sediment yield of the entire WDB catchment at the grounding zone equals the erosion rate, implying that nothing would accumulate there. That should be clarified.

*Based on reviewer comments, we modified our approach. In the revision, we now use the Bart and Tulaczyk (2020) yield (0.7 ± 0.21 mm yr$^{-1}$ for the Bindschadler Ice Stream) for only catchments entirely within West Antarctica. Gravity and aerogeophysical data from the WAIS suggests that the rift is primarily underlain by sedimentary rock (Wilson and Luyendyk, 2006; Tinto et al., 2019). This is only ground-truthed in a few locations where hot-water drilling through the ice accessed seafloor sediments that contained Miocene and Pliocene diatoms (Scherer et al., 1988). Seismic and multibeam data from the continental shelf show that sedimentary rocks dominate with minor areas of basement exposures known from the inner reaches of the Glomar Challenger Basin (Anderson and Bartek, 1992). We follow reviewer 2's recommendation to apply a 30% smaller yield (0.49 ± 0.21 mm yr$^{-1}$) for East Antarctic catchment areas because less-erodible basement rock that is expected to characterize the older East Antarctic craton (Li et al., 2020; Greenwood et al., 2021). For the GZWs in Glomar Challenger Basin, we applied the sediment yield of 0.7 ± 0.21 mm yr$^{-1}$ for the parts of the drainage area in Western Antarctica (Kamb, Whillans, and Mercer) and the sediment yield of 0.49 ± 0.21 mm yr$^{-1}$ for the parts of the drainage area in East Antarctica. This simplistic approach concerning the subglacial geology is warranted because there is too little information to make a detailed characterization of differences. We note that the Bart and Tulaczyk (2020) yield for the BIS is an average for a large percentage of West Antarctic catchment. The BIS catchment represents ~1/4 of the modern grounded area of the WAIS. We know of no data showing that either its subglacial geology or topography other West Antarctica are markedly different than that for the BIS catchment. On this basis, we favor the view that differences in neither subglacial geology nor topography would cause major changes in yield on West Antarctica. Please read the revised section 2.3 for more information on the new approach (Lines 90 - 100) and section 4.2 for the justification ( Lines 247 – 257).*

*Typically, the presence of meltwater contributes to higher sediment flux (Delaney and Adhikari, 2020). There is a significant hydrology below the WAIS today (Carter and Fricker, 2017). The modern network extends to all modern catchments suggesting that it contributes equally to the sediment fluxes of different catchments. The subglacial hydrology is partly related to atmospheric conditions, ice thickness, geothermal flux and ice flow (Simkins et al., 2023). These factors would have been different during the LGM and post-LGM. It is not known whether a similar network existed in the past. However, there is no evidence of either abundant meltwater*

*during the early deglacial that might have contributed to higher sediment fluxes or variations in meltwater discharge between catchments. Subglacial meltwater channels have been found on the middle shelf of the Pennell Trough and JOIDES basins (Simkins et al., 2015). This is the only large drainage system that has been found in the paleo-record for the Ross Sea continental shelf. Overall, meltwater deposition does not appear to be a significant part of the post-glacial sediment section in Ross Sea (e.g., Domack et al., 1999). Data from Prothro et al. (2020) suggests that meltwater deposits in the Pennell trough are neither thick nor extensive. We find no evidence suggesting that the different Ross Sea catchments experienced variations in meltwater discharge that would have impacted sediment fluxes.*

*The revision acknowledges that the onset, duration, and termination of a grounding-line stillstand depends on several factors and boundary conditions. Those factors could have included topographic controls including the steepness of subglacial profile, and atmospheric or ocean forcing (Lines 247 - 257). An examination the of the specific causes of the stillstands is beyond the scope of our study. Instead, the key objective of our study is generating first-order estimates of stillstand durations so that they can be compared for and between individual Ross Sea ice streams.*

- Chapter 4.1.3 "Stillstand durations within individual troughs" (Lines 205–209): This really depends on both sediment supply to the grounding zone and vulnerability of individual ice stream troughs to external forcing. So small scale features are not necessarily indicative of minor stillstands. Please elaborate on that more to justify your approach.

  *Specifically, we refer to small-scale ridges and backstepping wedge features that have been used to track retreat after the larger stillstands. These features are interpreted as grounding zone features but are thinner than the resolution limit of our seismic data. However, they can be mapped using CHIRP and multibeam bathymetry (Lines 230 – 233).We only have evidence that a major reorganization in flow only affected two of the mapped GZWs in the interior of JOIDES Basin (Lines 110- 113). Otherwise, we assume that sediment flux did not significantly change as the deglacial progressed.*

- Chapter 4.2 "Post-LGM erosion rates in Ross Sea" (Lines 215–220): I think only considering the degree of glaciation as a control factor on erosion rates is too simplified. Not only different topographies of those troughs but also subglacial geological variations along their axes may have controlled erosion rates by quite a lot, even though those troughs were exposed to the same

cold climate. This should at least be discussed as it may lead to further uncertainties for your calculations.

*Please see my reply to the first comment. We have added text discussing this in section 4.2 from (Lines 247 - 257).*

- Chapter 4.3 "A staggered post-LGM retreat of WAIS grounding lines in Ross Sea" (Lines 248–254): As already mentioned above, also the effect of subglacial geology should be discussed in more detail. Subglacial geological variations may be expressed as topography on the past ice sheet bed but do not necessarily have to but still may affect stillstand durations quite significantly (e.g., Dowdeswell & Fugelli, 2012, *GSA Bulletin*; Klages et al., 2014, *QSR*; Klages et al., 2015, *Geomorphology*). Does seismic information from within those troughs in your study area reveal sections of potentially indurated sediments (e.g., bedrock) or even basement cropping out at the former ice sheet bed? If yes that should be considered as an additional factor for inducing retreat deceleration, potential stillstand, and subsequent GZW accumulation.

*The majority of eastern Ross Sea (Whales Deep Basin and Little America Basin) is underlain by previously deposited subglacial sediment. The interior of Glomar Challenger Basin in central Ross Sea has exposed bedrock to the northwest of GZW j on Figure 4 that has been mapped from minor drumlin features on multibeam bathymetry. These outcrops are a relatively small portion of the drainage area of the eastern Ross Sea. The paleo-catchment area of the Whales Deep Basin likely contains a similar heterogeneity of basement and sedimentary outcrops. Thus, the average yield of $0.7 \pm 0.21$ mm yr$^{-1}$ sediment yield is probably appropriate.*

*The majority of western Ross Sea is underlain by previously deposited subglacial sediment on the outer shelf. Closer to Ross Ice Shelf, there is a higher presence of crystalline bedrock and indurated sedimentary bedrock (Greenwood et al., 2021). The upstream drainage areas in the Transantarctic Mountains also have crystalline bedrock and indurated sedimentary rock from outcrop data (Li et al., 2020). Thus, the average yield of $0.49 \pm 0.21$ mm yr$^{-1}$ sediment yield is probably appropriate.*

*For the GZWs in Glomar Challenger Basin, we applied the sediment yield of $0.7 \pm 0.21$ mm yr$^{-1}$ for the parts of the drainage area in Western Antarctica (Kamb, Whillans, and Mercer) and the sediment yield of $0.49 \pm 0.21$ mm yr$^{-1}$ for the parts of the drainage area in East Antarctica.*

*The variation in subglacial geology across Ross Sea is important and could have led to some of the differences in stillstand durations. We have added text discussing this in section 4.2 from (Lines 247 - 265).*

Minor

***Note: The reviewer comments are presented as previously posted in normal black font. Our responses appear below each reviewer comment italicized. All line numbers listed in responses to comments refer to the version with accepted changes.***

- Lines 10–11: Only by mapping GZWs you cannot say anything about stillstand durations. Please rephrase!

  *We have rephrased the statement (Lines 10 – 11)*

- Line 15: Please delete the "radiocarbon" in "…at least five thousand radiocarbon years." For referring to a period, it doesn't really matter if you refer to years, radiocarbon years, or calibrated years before present, etc. since you do not compare your age to anything. This is only necessary if you refer to an age and to elaborate more on the radiocarbon age and the calibrated age in comparison.

  *We have made the change (Line 15)*

- Figure 1: Please provide geographic context to both the Antarctic overview map (e.g., EAIS, WAIS, APIS, ocean basins, South Pole, etc.) but also your RSE map (names of coasts/lands, ocean basins, etc.). Also the depth scale needs to be more precise – two annotations at -40 and -1,500 mbsl is not enough for referring to the map. Cite the paper related to the IBCSO map.

  *We have made the requested edits to Figure 1 including more text in the caption (Lines 59 – 66)*

- Lines 91–92: Define sediment yield here. Yield at the grounding zone?

*We have added the definition (Line 88 - 89)*

- Figure 2: Same as for Fig. 1 here – please label geographic features for proper orientation and reference. Label the different catchments also within the figure for quicker reference. Lighter and darker shades for highlighting present and paleo drainage areas are really hard to differentiate – please change. And either change the depth scale as suggested for Fig. 1 or take out entirely because you cannot see much shelf bathymetry anyway.

  *We have made the requested edits to Figure 2 for better readability (Lines 117 – 125)*

- Line 119: Please clarify here. The seafloor beneath the Ross Ice Shelf could not be investigated in the scope of your study but generally it could be studied with e.g., vibroseismic methods.

  *We have added clarification (Lines 129 - 130)*

- Lines 126–127: The trough actually extends all the way to the modern GL and beyond towards the interior. Therefore, rather say "…three GZWs on the inner continental shelf proximal to the modern ice shelf edge."

  *We have made the change (Lines 133 -136)*

- Especially line 140 but also 191 and throughout: Reduce the number of abbreviations/acronyms. It's really confusing and distracting for the reader. Abbreviating "middle" and "outer continental shelf" for example is not necessary and also not common at all as it is referring to a general geomorphological feature.

  *We have reduced the use of acronyms throughout the entire text*

- Figure 3: Organize the three panels of the figure next to each other. As already mentioned for the other figures – please label geographic features for orientation and reference. It's also really hard for me to make out differences from the three plots. Maybe zoom in to specific locations to make that clearer. Same issues regarding the depth scale as for the other figures – needs to be more detailed.

  *We have adjusted this figure and moved it to supplemental figure 2 based on the feedback from Reviewer #2*

- Figure 4: Labels a–q need to be larger in the figure. Again: add general geographic names. Thickness scale needs to be more detailed. Depth scale for bathymetry is missing. Prominent troughs could be labelled in figure. Add reference to IBCSO.

  *We have adjusted this figure as requested (Lines 163 – 168)*

- Lines 262–263: This comment is a little bit in contrast to what you said earlier about initial retreat, i.e., a rather uniform response of the ice margin. Please clarify!

  *We state in the text that our data do not support the hypothesis that retreat from the LGM positions in the Ross Sea occurred in a uniform response. We have clarified this in the text (Lines 294 – 298).*

**Technical corrections**

***Note: The reviewer comments are presented as previously posted in normal black font. Our responses appear below each reviewer comment italicized. All line numbers listed in responses to comments refer to the version with accepted changes.***

- Title: I'm a little tripped up by the term "staggered" as it is not commonly used within the community. Rather use "episodic" or "stepwise", etc. You can also

be more specific about your study area, i.e., "across the Ross Sea Embayment shelf" or similar. Suggestion: "Episodic post-LGM grounding line retreat across the Ross Sea Embayment shelf, Antarctica".

- Line 8: Maybe avoid the term "greater" and write something like "The inland retreat of the WAIS in the Ross Sea sector was more significant than…".

   *We have made the change (Line 8)*

- Line 15: Avoid "progressed" and maybe write "…shorter stillstands throughout the deglacial". But as I said in my major comments…can we really be sure that they were shorter? You don't have age constraints and just by the size of a wedge it's risky to infer stillstand durations. Maybe the sediment supply was just a lot smaller.

   *We have made the change (Lines 15 – 16)*

- Line 16: "Over the course of the post-LGM" sounds weird. Suggestion: "…subsequent to the LGM" or "…throughout the deglacial".

   *We have made the change (Line 16)*

- Line 24: Define the acronym here and provide duration of the LGM in Antarctica in "cal. ka BP". You can delete "Antarctica in that sentence. And specify the location – "western Ross Sea" is too imprecise since the Ross Sea extends well into the deep ocean. Say "western Ross Sea Embayment" or similar. Same for referring to the eastern part.

   *We have adjusted the statements (Lines 24 - 25)*

- Line 30: Avoid mentioning "trough" twice and write "…foredeepened Drygalski Trough, JOIDES Basin, and Pennell Trough…".

   *We have made the change (Lines 29 – 30)*

- Lines 33–34: Rather write "During post-LGM ice sheet retreat, the GL retreat paused within the outer part of the GCB…".

  *We have made the change (Lines 33 – 34)*

- Line 51: Get rid of the second "regional" in that sentence.

  *We have made the change (Line 50)*

- Line 56: Rather write "The seismic profiles were interpreted…".

  *We have made the change (Lines 55)*

- Line 67: This sounds like that your seismic interpretation is of LGM age. Therefore, rather clarify and write e.g., "Seismic interpretation and isopach mapping of (post)-LGM grounding zone wedges".

  *We have made the change (Line 67)*

- Line 89: Add "as a basic parameter" between "used" and "to".

  *We have made the change (Line 87)*

- Line 91: Replace "concerning" with "to calculate". Add "for every ice stream" after "paleo-drainage area".

  *We have made the change (Lines 89)*

- Line 116: Change to "Seismically-resolvable GZWs on the Ross Sea Embayment shelf".

*We have made the change (Line 127)*

- Line 121: What do you mean by "GZWs have seafloor exposures"? That they were mapped and identified by multibeam bathymetry surveys? Please clarify!

  *We have removed the confusing statement*

- Line 123: What do you mean by "inner reaches of the middle continental shelf"? The transition from the inner to the mid shelf? Generally, try to avoid "inner reaches" here and throughout and change to "inner continental shelf" or similar.

  *The definition of Ross Sea we use here has the entire inner continental shelf covered by the Ross Ice Shelf which is why we use the term to refer to the innermost middle shelf*

- Line 178: Again, to me "Ross Sea" is too general. Specify to "Ross Sea Embayment shelf" or similar.

  *We have made the change (Line 193)*

- Line 180: Depending on what you mean get either rid of the "s" in "GZWs" or in "suggests".

  *We have edited the sentence (Lines 195-196)*

- Line 181: MCS as acronym was already introduced further up. But as I suggested above, it's kind of weird and not necessary to abbreviate general features such as shelf sections.

  *We have removed the abbreviation from the text*

- Lines 181–183: Please change to "GZWs on the RSE shelf are generally larger than those on other Antarctic continental shelves."

  *We have made the change (Lines 200-201)*

- Line 184: Change to "sedimentary bedrock".

  *We have made the change to sedimentary bedrock (Line 203)*

- Line 188: Change chapter title to "Variable stillstand durations between troughs".

  *We have made the change to the chapter title (Line 210)*

- Line 189: Specify to "Our data suggest that GL stillstands on the RSE shelf were of millennial to centennial durations."

  *We have made the change (Line 211)*

- Line 192: Change to "three millennia".

  *We have removed this sentence from the text*

- Line 194: Delete "the" before "millennial".

  *We have removed this sentence from the text*

- Line 195: Rephrase "would have had to have".

  *We have removed this sentence from the text*

- Table 5: What do you mean by "stratigraphic superpositions" here. Specify in caption or main text.

  *We have removed this table from the manuscript to focus on the approach mentioned in the first major comment*

- Line 202: Here you write "super-position" but "superposition" in Table 5. Decide for either one. And what do you exactly mean by "superposition"? That if you consider the seismic stratigraphy of the entire shelf that the outer shelf GZWs must be the oldest? Please clarify.

  *We have removed this table from the manuscript to focus on the approach mentioned in the first major comment*

- Line 204: As above – clarify if you mean transition from inner to mid shelf. Avoid acronym.

  *The acronym has been removed*

- Line 205: Please also cite more recent literature here, e.g., using data constraints or modeling evidence.

  *We have added another reference to the sentence, Joughin and Alley (2011) (Line 228 - 230)*

- Line 206: Change to "RSE troughs".

  *We have adjusted the sentence (Line 231)*

- Line 210: Delete "on the" in the chapter title.

  *We have made the change (Line 234)*

- Lines 214–215: Change to "…regional climate and associated precipitation, and the presence/absence of meltwater."

  *We have made the change (Line 238 -240)*

- Line 217: Replace "100%" with "entirely".

  *We have made the change (Line 241)*

- Line 224: Replace "from trough to trough" with "in between troughs".

  *We have removed this sentence*

- Chapter 4.3: Are all those ages really radiocarbon ages or were they calibrated? If calibrated, write "cal. ka BP". If some of them are radiocarbon ages and some were calibrated, you cannot relate them to each other here.

  *All ages discussed are calibrated. Omission of cal yr BP was an error (Line 267)*

- Lines 245–246: Change to "…in an unsteady episodic retreat style within individual troughs".

  *We have made the change (Line 279 - 280)*

- Line 256–257: Replace "from trough to trough" with "between troughs".

  *We have made the change (Lines 290 - 291)*

- Line 282: Same here.

*We have made the change (Line 320)*

**#2 Referee comment for preprint manuscript egusphere-2023-1397 "The staggered retreat of grounded ice in Ross Sea, Antarctica since the LGM" by Matthew A. Danielson and Philip J. Bart**

This manuscript explores the possible durations of grounding zone stillstands across the Ross Sea during post-LGM. To do this, the authors calculate the volumes of major grounding zone wedges using seismic data, and divide by sediment flux, which is derived from the product of the drainage area of the catchment and a value for sediment yield that was pulled from another Ross Sea publication. The results of the manuscript bolster the Antarctic research community's ever-increasing agreement that the Ross Sea's ice streams retreated asynchronously after the Last Glacial Maximum, first from the deep troughs, with grounded ice persisting on shallow banks for some time after.

The concept of this paper is interesting and the results could be valuable to the ice sheet modeling community; however, there are some issues in the execution of the study that should be considered, mostly related to the need for further exploration of boundary conditions. The authors have, understandably, had to make some assumptions for their calculations. It is fine to do this, but for a study that is so heavily dependent on assumed numerical boundary conditions, there should be more direct exploration of the range of possible values. After some additional analyses that showcase the breadth of possible grounding durations, these findings will be a valuable contribution to Antarctic science across disciplines.

**Major comments**

*Note: The reviewer comments are presented as previously posted in normal black font. Our responses appear below each reviewer comment italicized. All line numbers listed in responses to comments refer to the version with accepted changes.*

Firstly, the value for sediment yield that was chosen for the calculations was taken from a study of the age-constrained grounding zone wedge complex in Whales Deep Basin in the eastern Ross Sea. The authors point out that the catchment for the ice stream that once flowed through this trough has a largely sedimentary bed. The authors also point out that the catchment area for the ice flowing from the East Antarctic Ice Sheet into the western Ross Sea is floored by mostly crystalline bedrock, which can generate a 30% lower sediment yield than a sedimentary bed. This is a significant amount, yet it is dismissed. I would recommend the authors conduct another set of calculations using a 30% lower sediment yield, if not for the entire Ross Sea, then at least the western Ross Sea. They can then provide readers with a range of possible durations of grounding zone stillstands at each study location.

> *We agree that assumption of a single sediment yield across the entire Ross Sea is a major uncertainty. The focus on the yield from the Whales Deep Basin study (Bart and Tulacyzk, 2020) is due to the age-constrained grounding zone wedge in that area. In the revision, we now use the Bart and Tulaczyk (2020) yield (0.7 ± 0.21 mm yr$^{-1}$ for the Bindschadler Ice Stream) for only the eastern Ross Sea catchments of Little America Basin and Whales Deep Basin. We have adopted your recommendation to apply a 30% smaller yield (0.49 ± 0.21 mm yr$^{-1}$) for the western Ross Sea troughs of Pennell Trough, JOIDES Basin and Drygalski Trough . This is due to the presence of less erodible crystalline and sedimentary bedrock on the western Ross Sea shelf as well as in the upstream drainage area in the Transantarctic Mountains (Li et al., 2020; Greenwood et al., 2021). A 30% lower sediment yield is expected for crystalline bedrock (Schlunegger et al., 2001). For the GZWs in Glomar Challenger Basin, we applied the sediment yield of 0.7 ± 0.21 mm yr$^{-1}$ for the parts of the drainage area in Western Antarctica (Kamb, Whillans, and Mercer) and the sediment yield of 0.49 ± 0.21 mm yr$^{-1}$ for the parts of the drainage area in East Antarctica (Bart and Owolana, 2012). The text and tables have been adjusted to reflect the recalculation of the western Ross Sea troughs using the smaller sediment yield value. The new approach can be seen in section 2.3 of the methods (Lines 90 – 100) Further justification for the approach is in section 4.2 in the discussion (Lines 247 – 257).*

Another issue needing clarification is the approach for choosing particular catchment areas for the calculations. For instance, it isn't clear how the catchment area is divided between JOIDES and Pennell basins. Is the modern-day "brown" drainage area essentially halved for each, or is there some more nuanced reasoning on how the area Is partitioned into the two distributaries? It would be good to insert the values for each of the colored catchment areas as annotations on Figure 2, if appropriate, along with some sort of visual representation of how the JOIDES and Pennell areas were split.

> *The upstream catchment area for JOIDES and Pennell was divided evenly between the two distributaries. The total catchment upstream of the bifurcation was halved for flux*

*calculations in those two basins (Lines 107 - 109). We have edited the figure and its caption to make this clearer including adding a visual representation of the flow divide (Lines 117 – 125).We have added the areas of the catchment areas as annotations on Figure 2 for additional clarity.*

On a related note, there is an underlying assumption here that there was no reorganization of flow during retreat. Yet, Greenwood et al., 2018 (Nat. Comm.) show clear geomorphologic evidence of ice retreating from what appears to be the Danielson and Bart "JOIDES inner reach" GZWs toward Mawson Glacier, which is located north of Ross Island and flows through the Transantarctic Mountains. As shown in Greenwood et al., the catchment area could be quite different than what is used for calculations in this paper, as it would involve a majority of the flow coming from the pink drainage area in Fig. 2, not the brown. Perhaps the JOIDES inner reach calculations should be supplemented with another test using a drainage area that corresponds to this reorganization.

*Reorganization of flow occurred during the time of the JOIDES inner reach GZW deposition and retreat from these locations proceeded towards the Transantarctic Mountains. We have recalculated the two JOIDES inner reach GZWs with catchment areas that reflect reorganized flow from the David Glacier using the Greenwood et al., 2018 (Nat. Comm.) paper as a reference. These values are present in the tables and in section 2.3 of the methods (Lines 110 – 113).*

**Minor comments**

***Note: The reviewer comments are presented as previously posted in normal black font. Our responses appear below each reviewer comment italicized. All line numbers listed in responses to comments refer to the version with accepted changes.***

As seismic interpretation is so crucial to the outcome of these calculations, it might be prudent to include a figure demonstrating a before and after interpretation of a seismic line in the primary manuscript, so that readers can see how a GZW might be recognized in seismic.

*We have incorporated your other suggestions and Figure 3 has been replaced with the requested seismic figure. We have taken one of the seismic lines that was originally displayed in Supplemental Figure 2 and to make a figure with no interpretations and with*

*interpretations present to show a GZW interpretation from seismic data for GCB (Lines 155-160)*

Supplemental figure 2 is far too small. It is impossible to read the writing when printed on standard-sized paper.

> *We have changed Supplemental Figure 2 from its original version. We have taken one of the seismic lines for the new Figure 3. Please see the above and below points*

If I am being honest, the three maps in Fig. 3 all look identical. I have to squint really hard to spot a couple of contour lines that might be slightly different, and the colored depth scale bar in the legend is so condensed that I'm not gaining any information from it either. Unless there is a way to make the differences between the maps very clear, then I suggest removing this figure. I think the manuscript would be fine without it.

> *We stand by the utility of this figure for showcasing the regional changes in the Ross Sea bathymetry since the LGM. We have made this figure a supplemental figure and edited it to be clearer. It is now Supplemental Figure 2 and the original Supplemental Figure 2 has been modified to become Figure 3.*

Be careful not to exclude the EAIS from the narrative, as it is a significant contributor to the Ross Sea and delivered grounded ice to a large percentage of the Ross seafloor during more glaciated times, just like the WAIS. I noticed the EAIS was left out of the abstract and a few other places where it would be relevant.

> *We have added references to the EAIS in the abstract and throughout the text. We agree that it is a significant contributor to the Ross Sea and experienced expanded flow into western Ross Sea during the LGM.*

**Stylistic comments**

***Note: The reviewer comments are presented as previously posted in normal black font. Our responses appear below each reviewer comment italicized. All line numbers listed in responses to comments refer to the version with accepted changes.***

Some sections of this manuscript read very robotically, so just be on the lookout for areas where you could add transition words, vary sentence structure, and write in the active voice instead of passive. The abstract, in particular, should be revised with this in mind.

The Introduction needs some work in order to establish a narrative and show the broader research community why your work could be useful to them, so they will keep reading. As it currently stands, the Introduction launches immediately into Background information with no lead-in. The Introduction would benefit from a more deliberate structure. A suggested format: first paragraph that introduces the "Big Picture"—the reasons why Antarctica, the Ross Sea, and their geologic history are important. The second paragraph (a shortened version of your Lines 1 – 38) could give more niche background information that sets up the specific problem you investigate. The specific problem and previous attempts to solve it would be outlined in the third paragraph— perhaps Lines 45 – 47, followed by 38 – 43 and some explanation of what remains to be learned. A final, fourth paragraph could summarize what you do to solve it and could include Lines 43 – 45.

Consider expanding the Conclusion section (it is currently a little too perfunctory) to give space for suggestions of future work or returning to any "big picture" ideas from the introduction. Talk about the value of this study to the broader scientific community. Some transition words between thoughts could also improve the narrative.

*We have expanded the conclusion based on the feedback (Lines 315 – 325)*

It is a little jarring to read Ross Sea without "the" in front of it. I wouldn't directly refer to the Caspian Sea or the Atlantic Ocean without the "the" in front of them, unless I'm treating them as adjectives (e.g., "We analyzed Atlantic Ocean sediment"), so it is strange to treat the Ross Sea this way. Do consider modifying the title, abstract, and other instances of this wording.

*Thank you for the suggestion. We have made the correction throughout the text*

**Line edits**

*Note: All line numbers listed in responses to comments refer to the version with accepted changes*

Line 8: "greater" is an odd word choice—maybe say "more significant" and change "for" to "in."

> *We have made the correction (Line 8)*

Line 24 – 25: change to "outermost" and add "the" before "western" and "eastern". Add a comma after (wRS) and another after "Antarctica"

> *We have made the correction (Lines 24 – 25)*

Line 25: Remove "In other words"

> *We have made the correction (Line 25)*

Line 26: Remove "At that time"

> *We have made the correction (Line 26)*

Line 27: Add a comma after "broad" and remove "and" so that it says, "broad, foredeepened troughs"

> *We have made the correction (Line 27)*

Line 31: change to "backfilled"

> *We have made the correction (Line 31)*

Line 35: Add a comma after (LAB)

> *We have made the correction (Line 35)*

Line 44: add "the" before Ross Sea

*We have made the correction (Line 44)*

Line 45 – 46: Remove "the Ross Sea-wide" and add "in the Ross Sea" after "retreat"

*We have made the correction (Lines 45 – 46)*

Line 55: add a comma after "ages"

*We have made the correction (Line 54)*

Line 56: remove "the" before Petrel

*We have made the correction (Line 56)*

Line 70 – 74: consolidate these sentences to remove redundancies around the seafloor reflection and the unconformities associated with the GZWs.

*We have consolidated the sentences (Lines 70 – 73)*

Line 77: change "the Petrel software" to "Petrel" and add a comma after "meters" and remove the one after "primary input"

*We have made the correction ( Line 74)*

Line 81 – 82: join these sentences so it says "plotted in Petrel and interpolated to create depth and velocity maps"

*We have made the correction (Lines 79 - 81)*

Line 83 – 84: add "s" after "millisecond" and "meter"

*We have made the correction (Line 81 – 82)*

Line 88: remove "the" before "QGIS"

*We have made the correction (Line 86)*

Line 90: change "produce" to "product"

*We have made the correction (Line 88)*

Line 98: For this and other similar instances, I recommend simply writing out "outer continental shelf." Too many acronyms complicates this paper more than it needs to be, and it isn't customary to do that for unofficial place names anyway. I found myself wondering what bank or trough was abbreviated as "OCS" because I forgot what it meant.

*We have removed the acronyms across the manuscript*

Line 102: remove "the" before "David" and do capitalize "Glacier"

*We have made the correction (Line 109)*

Line 125: change "two" to "a" and add "two" after "and"

*We have made the correction (Lines 135 – 136)*

Line 127: "(Table 2; Fig. 4)"

*We have made the correction (Line 138)*

Line 128: what is meant by "define part of the banks"?

> *Part of the outer shelf deposition of the Hayes and Houtz banks occurred during the LGM (Line 139)*

Line 174: specify in Table 4 caption that the durations are measured in years

> *We have made the correction (Line 220 - 225)*

Line 180: edit to "The calculated durations suggest that the largest GZWs in the Ross Sea had stillstands lasting up to a few millennia"

> *We have made the correction (Lines 197 – 198)*

Line 181: give a little more detail about these radiocarbon dates—what range of ages, what material, what facies?

> *We have made the correction. "This general assessment is strongly supported by radiocarbon dates using benthic foraminifera collected in till of the WDB middle continental shelf stillstand (Bart et al., 2018). The dates suggest that the stillstand had begun by 14.7 ± 0.4 cal kyr BP before retreating by 11.5 ± 0.3 cal kyr BP." (Lines 198 – 200)*

Line 192: change "millennium" to "millennia"

> *We have reworked this section so this sentence is no longer present*

Line 194: remove "the" before "millennial" and add "have" before "been outside"

> *We have reworked this section so this sentence is no longer present*

Line 203: "data" is always plural, so change "suggests" to "suggest"

*We have made the correction (Line 226 - 227)*

Line 205: "predicts"

*We have made the correction (Line 229)*

Line 212: "ice streams"

*We have made the correction (Line 236)*

Line 217: remove "100%" and add a more qualitative word like "completely". Also add a comma after "grounded ice"

*We have made the correction (Line 241)*

Line 226: change "Antarctica" to "Antarctic"

*We have made the correction (Line 258)*

Line 234 – 236: are these ages calibrated? They need the correct notation. If uncalibrated, they shouldn't be compared to the calibrated ages. Also, for consistency, change the 8715 cal yr BP mentioned later in this paragraph to 8.7 cal kyr BP.

*We have made the corrections (Line 267 and Line 276)*

Line 248: change "small" to "low"

*We have made the correction (Line 282)*

Line 259: if this is the first mention of MWPs, it should be spelled out

*We have made the correction (Line 294)*

Line 262: change "current" to "modern" or "contemporary"

*We have made the correction (Line 297)*

Line 263: remove "style"

*We have made the correction (Line 298)*

Line 283 – 284: Perhaps say something similar to "…multiple episodes of small amplitude sea-level rise rather than a rapid increase in sea level from synchronous retreat"

*We have made the correction (Line 322-24)*

Line 284: change "ground" to "grounding"

*We have made the correction (Line 324)*

Anderson, J. B., and Bartek, L. R., 1992, Cenozoic Glacial History of the Ross Sea Revealed by
    Intermediate Resolution Seismic Reflection Data Combined with Drill Site Information,
    The Antarctic Paleoenvironment: A Perspective on Global Change: Part One, p. 231-264.

Bart, P. J., and Owolana, B., 2012, On the duration of West Antarctic Ice Sheet grounding events
    in Ross Sea during the Quaternary: Quaternary Science Reviews, v. 47, p. 101-115,
    10.1016/j.quascirev.2012.04.023.

Carter, S. P., and Fricker, H. A., 2017, The supply of subglacial meltwater to the grounding line
    of the Siple Coast, West Antarctica: Annals of Glaciology, v. 53, no. 60, p. 267-280,
    10.3189/2012AoG60A119.

Delaney, I., and Adhikari, S., 2020, Increased subglacial sediment discharge in a warming
    climate: Consideration of ice dynamics, glacial erosion, and fluvial sediment transport:
    Geophysical Research Letters, v. 47, no. 7, 10.1029/2019gl085672.

Greenwood, S. L., Simkins, L. M., Winsborrow, M. C. M., and Bjarnadóttir, L. R., 2021,
    Exceptions to bed-controlled ice sheet flow and retreat
from glaciated continental margins worldwide: Science Advances, v. 7, p. 1 - 12.

Li, X., Zattin, M., and Olivetti, V., 2020, Apatite Fission Track Signatures of the Ross Sea Ice Flows
    During the Last Glacial Maximum: Geochemistry, Geophysics, Geosystems, v. 21, no. 10,
    10.1029/2019gc008749.

Scherer, R. P., Harwood, D. M., Ishman, S., and Webb, P. N., 1988, Micropaleontological analysis
    of sediments from the Crary Ice Rise,Ross Ice Shelf: ANTARCTIC JOURNAL.

Schlunegger, F., Melzer, J., and Tucker, G., 2001, Climate, exposed source-rock lithologies,
    crustal uplift and surface erosion: a theoretical analysis calibrated with data from the
    Alps/North Alpine Foreland Basin system: International Journal of Earth Sciences, v. 90,
    no. 3, p. 484-499, 10.1007/s005310100174.

Simkins, L. M., Greenwood, S. L., Winsborrow, M. C. M., Bjarnadóttir, L. R., and Lepp, A. P.,
    2023, Advances in understanding subglacial meltwater drainage from past ice sheets:
    Annals of Glaciology, v. 63, no. 87-89, p. 83-87, 10.1017/aog.2023.16.

Tinto, K. J., Padman, L., Siddoway, C. S., Springer, S. R., Fricker, H. A., Das, I., Caratori Tontini, F.,
    Porter, D. F., Frearson, N. P., Howard, S. L., Siegfried, M. R., Mosbeux, C., Becker, M. K.,
    Bertinato, C., Boghosian, A., Brady, N., Burton, B. L., Chu, W., Cordero, S. I., Dhakal, T.,
    Dong, L., Gustafson, C. D., Keeshin, S., Locke, C., Lockett, A., O'Brien, G., Spergel, J. J.,
    Starke, S. E., Tankersley, M., Wearing, M. G., and Bell, R. E., 2019, Ross Ice Shelf
    response to climate driven by the tectonic imprint on seafloor bathymetry: Nature
    Geoscience, v. 12, no. 6, p. 441-449, 10.1038/s41561-019-0370-2.

Wilson, D. S., and Luyendyk, B. P., 2006, Bedrock platforms within the Ross Embayment, West
    Antarctica: Hypotheses for ice sheet history, wave erosion, Cenozoic extension, and
    thermal subsidence: Geochemistry, Geophysics, Geosystems, v. 7, no. 12, p. n/a-n/a,
    10.1029/2006gc001294.

---

## Author Response (AR2)

*Note: The reviewer comments are presented as previously posted in normal black font. Our responses appear below each reviewer comment italicized. All line numbers listed in responses to comments refer to the version with tracked changes.*

Reviewer #2

**Suggestions for revision or reasons for rejection**

The second version of this manuscript is much improved from the previous version I reviewed in terms of data presentation and narrative. I thank the authors for the edits to the figures—they have addressed a lot of questions that I had. The new version of the conclusion is very well done and adequately demonstrates the impact of this work on the scientific community.

I have two remaining minor concerns that may or may not involve recalculations. The first is regarding the catchment area that is being used for the GZWs (m and n) in the JOIDES inner reaches. At my previous recommendation, the authors have considered the ice flow reconfiguration during the later stages of grounded ice retreat in the southwestern Ross Sea, which is indicated by glacial landforms on the seafloor (Greenwood et al., 2018). The authors have adjusted the catchments for their JOIDES wedges to reflect the initial northward flow from the Byrd catchment, which fed the middle JOIDES GZW, followed by reorganization of JOIDES ice flow that caused the inner reaches to be sourced from the nearby outlet glaciers of the Transantarctic Mountains. For the inner reaches, it appears the authors have used only the David Glacier catchment area for the calculations, rather than the entire pink polygon representing all flow from the Transantarctics. This is good, because the small catchments north of David Glacier are irrelevant at these late stages of retreat. However, I don't think the authors incorporated the southernmost catchment represented by the pink polygon, which is important because it represents outlet glaciers that fed the readvance in southern JOIDES following the reorganization of flow. These southernmost glaciers, and in particular Mawson and Mackay Glacier, were arguably the greatest contributors to the readvance of ice and suppliers of GZW sediment in the inner reaches. In contrast, seafloor geomorphology indicates that David Glacier probably contributed but did not dominate at this time (Greenwood et al., 2018). Although it probably will not change the calculated values drastically, I do suggest the authors adjust their catchment area for the JOIDES inner reach wedges (m and n).

*The reorganized drainage area in the previous version uses the catchments from Mawson and Mackay Glacier in addition to David Glacier. The southernmost polygon in the pink region of Figure 2 contains the drainage areas of both Mawson and Mackay Glaciers based on the drainage area definitions from IMBIE 2016. We have clarified the confusing text in the manuscript (Lines 116 – 118).*

My second concern also involves the JOIDES inner reach GZWs (m and n). I only ask that the authors verify whether either of these GZWs have been shown in other recent publications. I suspect that GZW "m" is the same as the inner JOIDES GZW complex described in both Greenwood et al., 2018 and Simkins et al., 2017, but it is hard to tell in these figures. If it is the same GZW, then it is the one that was fed by the reorganized flow from the Transantarctic Mountains, as mentioned above and described in this manuscript. In my last review, I assumed that both "m" and "n" had the same catchment due to their proximity to each other, and this manuscript seems to make the same assumption in its revision. However, this may have been an oversight. I question the origin of "n." It may be possible that "n" was formed during original southward retreat into the Byrd catchment just prior to the reorganization, and "m" followed after the reorganization and readvance, as described in the aforementioned publications, thus making "n" the older of the two GZWs. The authors should look more closely at these GZWs and compare them to existing publications to verify or refute the claim that both catchments are to the west. Additionally, they should indicate which, if any, GZWs from previous publications are equivalent to the "m" and "n" wedges and are discussed in greater detail, so that their catchment choices will be further supported and readers may more easily synthesize this new work with prior work.

*GZW m is the same as the one identified in the Greenwood et al., 2018 and Simkins et al., 2017. We previously listed that as one of the GZWs that was described in earlier studies. A more detailed description of the previously identified GZWs is available in Appendix 1 and Supplemental Table 3 of the Supplemental.*

*After review, GZW n was most likely formed during original southward retreat into the Byrd catchment after retreat from the Pennell Middle Shelf GZW position. This deposition occurred prior to the reorganization that is responsible for GZW m. We have corrected the calculation to use the Byrd drainage area for this GZW (Table 1; Line 195). We have also added text in the discussion to clarify that GZW n was deposited first and GZW m was deposited after the reorganization. (Lines 271 – 274). GZW n was not identified in a previous study and is outside of the multibeam coverage used in Greenwood et al., 2018 and Simkins et al., 2017.*

Two minor technical corrections:

*We have made the change (Line 382)*

*We have made the change (Line 385)*